# Defining key concepts for mental state attribution

François Quesque, Ian Apperly, Renée Baillargeon, Simon Baron-Cohen, Cristina Becchio, Harold Bekkering, Daniel Bernstein, Maxime Bertoux, Geoffrey Bird, Henryk Bukowski, Pascal Burgmer, Peter Carruthers, Caroline Catmur, Isabel Dziobek, Nicholas Epley, Thorsten Michael Erle, Chris Frith, Uta Frith, Carl Michael Galang, Vittorio Gallese, Delphine Grynberg, Francesca Happé, Masahiro Hirai, Sara D. Hodges, Philipp Kanske, Mariska Kret, Claus Lamm, Jean Louis Nandrino, Sukhvinder Obhi, Sally Olderbak, Josef Perner, Yves Rossetti, Dana Schneider, Matthias Schurz, Tobias Schuwerk, Natalie Sebanz, Simone Shamay-Tsoory, Giorgia Silani, Shannon Spaulding, Andrew R. Todd, Evan Westra, Dan Zahavi & Marcel Brass

The terminology used in discussions on mental state attribution is extensive and lacks consistency. In the current paper, experts from various disciplines collaborate to introduce a shared set of concepts and make recommendations regarding future use.

Our daily social interactions crucially rely on our ability to understand what other people think, believe, feel, intend and perceive. Such mental states, unlike others' overt behaviors, are not directly observable. The last several decades have witnessed a growing interest in understanding the cognitive and neural bases of mental state attribution, both in order to gain a deeper understanding of social cognition broadly and to improve our understanding and treatment of clinical conditions characterized by differences in social interaction in particular. However, fifty years into social-cognitive research, the very structure of social cognition is still poorly understood. Empirical and theoretical progress in this research domain is largely impeded by the extremely heterogeneous taxonomy currently used to describe key constructs. Sometimes a specific term is used to describe different constructs (e.g., 'theory of mind'[1], 'empathy'[2]) and sometimes different terms are used to describe the same construct (e.g. 'theory of mind', 'mentalizing', 'mindreading'). Consequently, the literature on mental state attribution is currently blurred by dozens of terms, which are sometimes used as synonyms and sometimes to describe complementary concepts.

Interestingly, this lack of consensus is observed not only across different disciplines but also within certain disciplines in which umbrella terms at the conceptual level cover methodological variability (e.g. the same concept is used to refer to explicit beliefs ascription and emotional faces categorization). This heterogeneity leads to problems in the comparability and generalization of findings. Therefore, it would be beneficial if researchers interested in mental state attribution could rely on a common set of definitions of key constructs. Identifying the consensual heuristic meanings associated with the different terms in use in the literature would undoubtedly make the appreciation of the current theoretical challenges easier. A similar enterprise was successfully conducted in the field of emotions a decade ago[3]. This work revealed that while multiple competing definitions coexist, considerable agreements were reached on aspects of the definitions. Such initiatives clearly sharpen perspectives on the main issues

in the field and generate fruitful research programs on the most divisive aspects.

Of course, given different theoretical perspectives, it may be an impossible task to reach a perfect consensus for every construct in use. However, because there are currently no empirical studies investigating levels of agreement/disagreement among experts on definitions of socio-cognitive constructs, researchers must individually navigate the conceptual landscape to gain a sense of where such agreements/disagreements exist. This state of affairs not only slows scientific progress but also disproportionately affects early career researchers who are attempting to get a foothold in the field. In addition to these theoretical impacts, clinical practices are also affected by the absence of harmonization both for diagnosis purposes and for patients' monitoring. As such, it would be helpful to have a common lexicon of key constructs, with the amount of agreement easily available.

To initiate the development of a shared lexicon, FQ and MB engaged with experts in the field. Although the selection was not exhaustive, it encompasses a diverse array of disciplines, countries, and career stages (see Supplementary Table 1). Then, our panel of researchers collaboratively identified essential components of such definitions for each construct. In this process, we tried to avoid convergent use of terms (different terms referring to the same concept) and divergent use of terms (one term referring to different concepts). Importantly, such a pruning should be as consensual as possible to ensure that researchers commit to the outcome of this process. In Box 1, we present the product of this collaborative project – a common lexicon of mental state attribution terms developed in consultation with leading researchers from diverse relevant research fields (e.g. affective neuroscience, philosophy, social psychology).

## Choosing and discarding terms

Following the process outlined above and in more detail in the Supplementary Methods, we agreed on a set of most important terms. In Box 1, we outline the recommendation for whether the term should further be used and provide our consensually generated definition and percentage of agreement. The remaining terms referred to definitions that were already covered by more suitable alternatives. For example, the definitions obtained for 'cognitive empathy' largely overlap with the present definition of mentalizing. While introducing the concept might have had a clear justification two decades ago (i.e., "the cognitive component of empathy"), it now seems redundant and conceptually problematic. Accordingly, a significant

proportion of our panel were concerned about its intrinsic ambiguity and support this term to be discontinued (see Fig. 1).

Table 1 summarizes our recommendations for each of the discarded terms. It might appear surprising that for some terms (e.g., "empathic perspective-taking") we recommend favoring alternatives which, at first sight, might seem less lexically related (e.g., "mentalizing about affective states") than others (e.g., "empathy" or "perspective-taking"). The reason driving these decisions is that we focused on the conceptual idea underlying the panel's definitions of these terms, rather than on lexical proximity or alternative definitions. In this case, as "empathic perspective-taking" was defined as 'imagining how a target person feels', it appeared to be conceptually closer to our proposed definition of mentalizing than to those obtained for empathy (as its definition does not imply experiencing the emotional state of the targeted person) or perspective-taking (as its definition does not imply the adoption of the other perspective).

### Advantage of a common lexicon

The present collaborative initiative allowed us to summarize the current state of the discussions among specialists from various fields, ranging from affective neuroscience to philosophy. This lexicon offers a clear orientation for people entering the field and can act as a basis for further interdisciplinary discussion about the use of terms. Among the 45 researchers who accepted to take part on this project, 43 ultimately accepted to be listed as co-authors, which illustrates a general willingness to compromise from researchers. Despite the collaborative nature of our endeavor and the high range of agreements obtained, we acknowledge that complete consensus in

---

## Box 1 | The Lexicon of mental state attribution terms

**Mentalizing**

We defined 'mentalizing' as "*the ability to attribute mental states (e.g., knowledge, intentions, emotions, perception) to self and others*". Ninety percent of our expert panel agreed with this definition. Most of the experts used "mentalizing" as a strict synonym of "theory of mind" (66%) and of "mindreading" (61%), to refer to the broad ability to attribute mental states to others. As only 13% of the expert panel originally voted to no longer use this term in favor of a synonym and 27% stated that they would favor this term, Mentalizing has ultimately been selected as the most generic term to use when addressing the ability to attribute mental states.

**Theory of mind**

Two main definitions of "theory of mind" currently co-exist in the literature and were accordingly listed by our experts. One of these definitions refers to the ability to attribute mental states (i.e. mentalizing), whereas the other refers to a theory-specific term for the hypothesis that thinking about other people's mental states involves a set of concepts and principles about how these concepts interact. We recommend using "theory of mind" exclusively to refer to its second meaning and define it as "*The use of folk psychological knowledge and heuristics (e.g., "mental states are correlated with behaviors", "mental states differ between agents") to think about one's own and other people's mental states*". In this way, having a theory of mind would indicate one specific way (among multiple) to mentalize. 80% of the expert panel agreed with this definition. Importantly, this definition is congruent with the original use of "theory of mind" in ethology in which it was viewed as a theory that individuals hold and not as an ability[5].

**Empathy**

The theoretical heterogeneity associated with the term "empathy" has been widely discussed recently[2,6,7]. This is also reflected in the heterogeneity of definitions that we received from the expert panel. To limit synonyms as much as possible, and thus to clarify theoretical elaborations, we recommend adopting the following definition when speaking of empathy in the context of mental state attribution: "*the ability to experience others' affective states, while maintaining the distinction from one's own affective states*". Among our panel, 82.5% finally agreed with this definition. We recognize that the present recommendation might not convince all researchers, from all fields. However, we hope that the present collaborative work will at least help to prevent the use of "empathy" to refer to any kind of mental state attribution as less ambiguous alternatives are available (e.g. mentalizing about affective states).

**Perspective-taking**

We recommend defining perspective-taking as "*the process by which one represents others' mental states, by adopting their perspective*". Based on this definition, perspective taking would refer to a specific form of mentalizing. Importantly, perspective-taking can refer to a spatial (e.g., "my colleague think that I am working as they can't see my screen") and a temporal perspective (e.g., "tomorrow my colleague will be disappointed, if I don't prepare the meeting now"). 82,5% of our panel agreed with this definition.

**Visuo-spatial perspective-taking**

Contrasting with the other terms, initial definitions obtained from the expert panel were highly consensual and allowed us to define visuo-spatial perspective-taking as "*the process by which one represents a scene from another person's viewpoint, by adopting their perspective*". Eighty percent of our panel agreed with this definition.

**Level 1 Visuo-spatial perspective-taking**

Again, initial definitions obtained from the expert panel were largely consensual and allowed us to define level 1 visuo-spatial perspective-taking as "*the process by which one determines whether or not visual stimuli are in the sight of another person or not*". 84% of our panel agreed with this definition.

**Level 2 Visuo-spatial perspective-taking**

The definition of level 2 visuo-spatial perspective-taking (i.e., *the process by which one represents a scene from another person's viewpoint, by adopting their perspective*) largely overlaps with the general definition of visuo-spatial perspective-taking. This redundancy between definitions is widely acknowledged in the literature and would usually motivate us to discard the term. However, given that it is primarily used in combination with level 1 visuo-spatial perspective-taking, the term seems useful and does not lead to confusion. Therefore, our suggestion is to limit the use of these specific terms to situations where the distinction is needed (in the field of visuo-spatial perspective-taking) while speaking of visuo-spatial perspective-taking when no distinction is required. 79% of our panel agreed with this definition.

**Self-other distinction**

A high consensus was achieved among the definitions obtained from our expert panel. This allowed us to define self-other distinction as "*the process by which one distinguishes between self- and other-related representations (cognitive, affective, sensorimotor, etc.)*". 84% of our panel agreed with this definition.

---

**Fig. 1 | Proportion of our panel of researchers who agree for each term to be discontinued.** Terms are ordered from the most to the least frequently encountered (Proportion of our panel of researchers who stated having already encountered each term is available in more detail as Supplementary Fig. 1). Proportions are based on all responses available for each question.

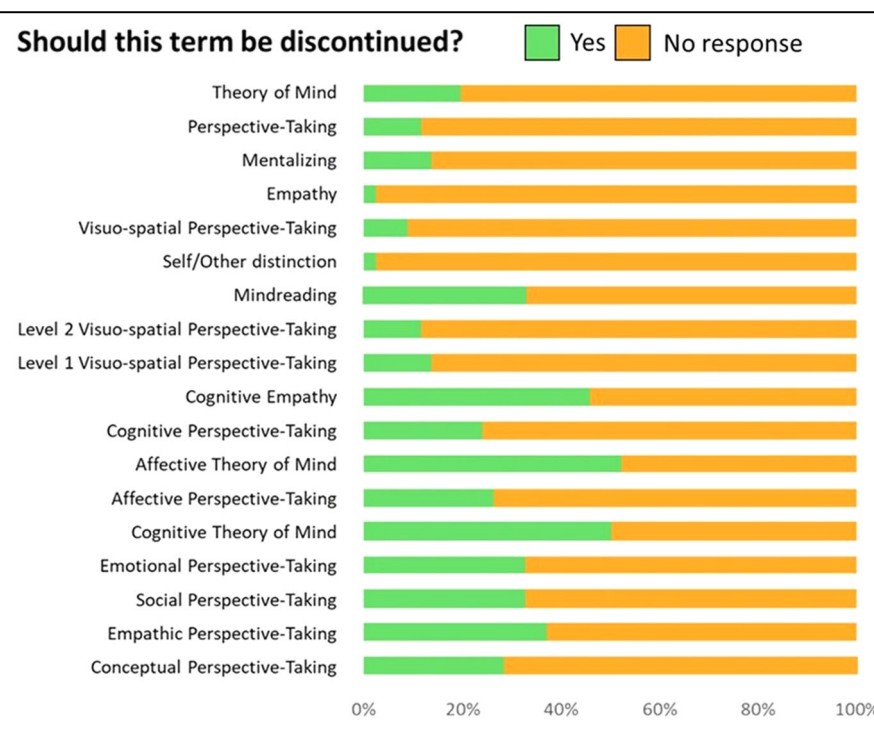

## Table 1 | Summary of our recommendations for the alternatives to each of the discarded terms

| Original terms | Suggested alternatives | Percentage of agreement |
|---|---|---|
| Mindreading | Mentalizing | 82% |
| Affective Theory of Mind | Mentalizing about affective states | 92% |
| Cognitive Theory of Mind | Mentalizing about beliefs N.B: or mentalizing about intentions/desires/etc., depending on the specific context. | 84.6% |
| Cognitive Empathy | Mentalizing | 85% |
| Social Perspective-taking | Perspective-taking | 90% |
| Affective Perspective-taking | Perspective-taking | 85% |
| Empathic Perspective-taking | Mentalizing about affective states | 83% |
| Emotional Perspective-taking | Mentalizing about affective states | 82% |
| Cognitive Perspective-taking | Mentalizing about beliefs N.B: or mentalizing about intentions/desires/etc., depending on the specific context. | 85% |
| Conceptual Perspective-taking | Mentalizing | 87% |

These recommendations are based on the consensual definitions obtained.

the field is an unlikely outcome. Accordingly, there is not a single term for which all researchers agreed on a definition. This also means that being listed as an author in the present manuscript does not indicate complete agreement with all definitions. Rather, authorship indicates a contribution to the collaborative effort to reach consensus.

In contrast to previous attempts of conceptual clarification in the domain, our approach is not based on a specific theoretical position or a specific disciplinary perspective. Rather, we relied on an eclectic approach, involving experts from various scientific fields with diverse theoretical backgrounds to acknowledge the diversity of perspectives, and to maximize the impact and scope of the consensual lexicon. Despite this heterogeneity, we reached a high consensus on most of the introduced terms, suggesting that disagreement in the literature partly originates from a lack of

communication among different research fields rather than from an in-principle inability to agree. However, we also must acknowledge that reaching a perfect consensus in such a large and heterogeneous group of researchers (as well as outside our panel in the context of future use) is illusory. We tried to account for this fact by reporting the amount of agreement for each definition. Furthermore, we carefully outlined the rationale behind the decision for each term. We are aware that the implementation of the lexicon we propose might encounter some resistance because it goes against long-standing traditions in the field. A term like "theory of mind", for example, has been used as a synonym for "mentalizing" over a long period of time. Therefore, discarding these terms or using them differently is challenging. However, we think that conventions and habits should not prevent us from trying to implement improvements.

## Moving forward

Although we have motivated the benefits of adopting a common lexicon in advancing scientific knowledge, it is nevertheless still likely that individual preferences for old terms will prevail simply due to habit or to the old term feeling more concise. For example, because "mentalizing about affective states" might be considered less concise than "affective mentalizing", it might be tempting to reintroduce old terms or even to generate novel combinations. Although we are sympathetic to such behavior, we suggest that this lexicon is a starting point for researchers to clean up conceptual clutter. Our goal is not to stifle theoretical and methodological innovations by setting in stone definitions of complex constructs. Nevertheless, if individual researchers have preferences for old or even novel terms to describe whatever phenomenon they are studying, then we would encourage them to state why the use of these terms is needed. Doing so not only allows individual researchers to express their creativity when studying these phenomena but also makes explicit to readers where and why they have deviated from the norms of the field. Such an approach may even lead to an update of the current lexicon as more researchers become convinced that the old or novel term better captures the phenomena.

The collaborative approach to develop a common lexicon proposed here constitutes a crucial first step towards better practices. This approach also has limitations that we want to address. First, despite a willingness to involve as many experts as possible, we acknowledge that our consortium might be biased and therefore it would be crucial to expand it in the next steps. Thus, we think that this initiative will have to be followed by other collective initiatives to ensure the representation of different cultures, as well as the different aspects of psychology and social neuroscience. A second limitation comes from the consensual approach itself, which might be problematic for defining terms. One might argue that searching for consensus prevents the development of more radical or ground-breaking ideas, which could eventually be suitable for significant scientific advances[4]. However, we would like to mention that past large-scale consensual projects have generated fruitful methodological progress by harmonizing researchers' practices. Independent of our consensual suggestions, the present work should act as a strong signal for exercising more caution in terminological selection when discussing findings originating from various paradigms. A third limitation stems from the formulation of the questions presented to our panel of experts. Using open-ended questions allowed us to capture the components associated with the different terms that were shared among our experts, but it also prevented us from reporting precise proportions of agreement for each of these components. Future work using more fine-grained evaluations will be necessary to quantify how much agreement any sub-components of each definition receive.

François Quesque [1]✉, Ian Apperly[2], Renée Baillargeon[3],
Simon Baron-Cohen [4], Cristina Becchio [5], Harold Bekkering[6],
Daniel Bernstein [7], Maxime Bertoux[8], Geoffrey Bird[9],
Henryk Bukowski[10], Pascal Burgmer[11], Peter Carruthers[12],
Caroline Catmur [13], Isabel Dziobek[14], Nicholas Epley[15],
Thorsten Michael Erle [16], Chris Frith [17], Uta Frith[18],
Carl Michael Galang[19], Vittorio Gallese [20,21], Delphine Grynberg[22,23],
Francesca Happé[24], Masahiro Hirai[25,26], Sara D. Hodges[27],
Philipp Kanske [28], Mariska Kret[29,30], Claus Lamm [31],
Jean Louis Nandrino[22,32], Sukhvinder Obhi[33], Sally Olderbak[34],
Josef Perner[35], Yves Rossetti [36], Dana Schneider[37],
Matthias Schurz[38,39], Tobias Schuwerk[40], Natalie Sebanz[41],
Simone Shamay-Tsoory [42], Giorgia Silani [43], Shannon Spaulding[44],
Andrew R. Todd [45], Evan Westra[46], Dan Zahavi[47] & Marcel Brass[19]✉

[1]Centre Ressource de Réhabilitation Psychosociale, CH Le Vinatier, Lyon, France. [2]School of Psychology University of Birmingham B15 2TT, Birmingham, UK. [3]Department of Psychology, University of Illinois, Urbana-Champaign, USA. [4]Autism Research Centre, Psychiatry Department, Cambridge University, Cambridge, UK. [5]Department of Neurology, Hamburg Center of Neuroscience, University Medical Center Hamburg-Eppendorf, Martinistraße 52, 20246 Hamburg, Germany. [6]Donders Centre for Cognition, Donders Institute for Brain, Cognition, and Behaviour, Radboud University Nijmegen, Nijmegen, the Netherlands. [7]Kwantlen Polytechnic University, Surrey BC V3W 2M8, Canada. [8]Lille Neuroscience & Cognition, Univ. Lille, Inserm, CHU Lille, LiCEND & DistALZ, Lille, France. [9]Dept of Experimental Psychology, University of Oxford, and School of Psychology, University of Birmingham, Birmingham, UK. [10]Psychological sciences research institute, Université catholique de Louvain, Louvain-La-Neuve, Belgium. [11]Center for Research on Self and Identity, School of Psychology, University of Southampton, Southampton SO17 1BJ, UK. [12]Department of Philosophy, University of Maryland, College Park, MD 20472, USA. [13]Department of Psychology, Institute of Psychiatry, Psychology, and Neuroscience, King's College London, London, UK. [14]Clinical Psychology of Social Interaction, Institute of Psychology, Humboldt-Universität zu Berlin, Berli, Germany. [15]University of Chicago Booth School of Business, Chicago, USA. [16]Department of Social Psychology, Tilburg University, Tilburg, the Netherlands. [17]Wellcome Centre for Human NeuroImaging, University College London, 12 Queen Square, London WC1N 3AR, UK. [18]University College London, Institute of Cognitive Neuroscience, London, UK. [19]Humboldt University of Berlin, Berlin School of Mind and Brain, Department of Psychology, Berlin, Germany. [20]Dept. of Medicine & Surgery, Unit of Neuroscience, University of Parma Via Volturno, 39 43121 Parma, Italy. [21]Italian Academy for Advanced Studies in America, Columbia Univeristy, New York, USA. [22]Univ. Lille, CNRS, UMR 9193, SCALab - Sciences Cognitives et Sciences Affectives, F-59000 Lille, France. [23]Institut Universitaire de France, Paris, France. [24]Social, Genetic & Developmental Psychiatry Centre - Institute of Psychiatry, Psychology & Neuroscience - King's College London, London, UK. [25]Department of Cognitive and Psychological Sciences, Graduate School of Informatics, Nagoya University, Nagoya, Japan. [26]Furo-cho, Chikusa-ku, Nagoya 464-8601, Japan. [27]Department of Psychology, University of Oregon, Eugene, USA. [28]Clinical Psychology and Behavioral Neuroscience, Faculty of Psychology, Technische Universität Dresden, Chemnitzer Str. 46, 01187 Dresden, Germany. [29]Cognitive Psychology Unit, Institute of Psychology, Leiden University, Leiden, the Netherlands. [30]Leiden Institute for Brain and Cognition (LIBC), Leiden, the Netherlands. [31]Department of Cognition, Emotion, and Methods in Psychology, Faculty of Psychology, University of Vienna, Vienna, Austria. [32]Fondation Santé des étudiants de France (FSEF), Paris, France. [33]Department of Psychology, Neuroscience & Behaviour, McMaster University, 1280 Main St West Hamilton, Ontario, Canada L8S 4L8, Canada. [34]Institut für Therapieforschung, Munich, 175 Leopoldstraße, Munich 80804, Germany. [35]Centre for Cognitive Neuroscience & Department of Psychology, University of Salzburg, Salzburg, Austria. [36]Université Claude Bernard Lyon 1, CNRS, INSERM, Centre de Recherche en Neurosciences de Lyon CRNL U1028 UMR5292, Trajectoires F-69500 Bron, France. [37]Institute of Psychology, Friedrich Schiller University Jena, Jena, Germany. [38]Institute of Psychology, University of Innsbruck, Universitätsstraße 5-7, 6020 Innsbruck, Austria. [39]Digital Science Center (DiSC), University of Innsbruck, Innrain 15, 6020 Innsbruck, Austria. [40]Department of Psychology, Ludwig-Maximilians-Universität München, Munich, Germany. [41]Central European University PU, Department of Cognitive Science, Vienna, Austria. [42]Department of Psychology,

University of Haifa, Haifa, Israël. ⁴³Department of Clinical and Health Psychology, University of Vienna, Vienna, Austria. ⁴⁴Oklahoma State University, Department of Philosophy, Stillwater, USA. ⁴⁵Department of Psychology, University of California, Davis, 1 Shields Avenue, Davis, CA 95616, USA. ⁴⁶Department of Philosophy, Purdue University, Lafayette, USA. ⁴⁷Center for Subjectivity Research, University of Copenhagen, Copenhagen, Denmark.
✉e-mail: francois.quesque@gmail.com; marcel.brass@hu-berlin.de

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

## Author contributions

Conceptualization: François Quesque and Marcel Brass; Inviting expert panel as co-authors: François Quesque and Marcel Brass; Providing individual definitions and opinions: Ian Apperly, Renée Baillargeon, Simon Baron-Cohen, Cristina Becchio, Harold Bekkering, Daniel Bernstein, Maxime Bertoux, Geoffrey Bird, Marcel Brass, Henryk Bukowski, Pascal Burgmer, Peter Carruthers, Caroline Catmur, Isabel Dziobek, Nicholas Epley, Thorsten Michael Erle, Chris Frith, Uta Frith, Carl Michael Galang, Vittorio Gallese, Delphine Grynberg, Francesca Happé, Masahiro Hirai, Sara D. Hodges, Philipp Kanske, Mariska Kret, Claus Lamm, Jean Louis Nandrino, Sukhvinder Obhi, Sally Olderbak, Josef Perner, François Quesque, Yves Rossetti, Dana Schneider, Matthias Schurz, Tobias Schuwerk, Natalie Sebanz, Simone Shamay-Tsoory, Giorgia Silani, Shannon Spaulding, Andrew R. Todd, Evan Westra, Dan Zahavi ; Voting for recommendations and consensually generated definitions: Ian Apperly, Renée Baillargeon, Simon Baron-Cohen, Cristina Becchio, Harold Bekkering, Daniel Bernstein, Maxime Bertoux, Geoffrey Bird, Marcel Brass, Henryk Bukowski, Pascal Burgmer, Peter Carruthers, Caroline Catmur, Isabel Dziobek, Nicholas Epley, Thorsten Michael Erle, Chris Frith, Uta Frith, Carl Michael Galang, Vittorio Gallese, Delphine Grynberg, Francesca Happé, Masahiro Hirai, Sara D. Hodges, Philipp Kanske, Mariska Kret, Claus Lamm, Jean Louis Nandrino, Sukhvinder Obhi, Sally Olderbak, Josef Perner, François Quesque, Yves Rossetti, Dana Schneider, Matthias Schurz, Tobias Schuwerk, Natalie Sebanz, Simone Shamay-Tsoory, Giorgia Silani, Shannon Spaulding, Andrew R. Todd, Evan Westra, Dan Zahavi; Writing—original draft: François Quesque and Marcel Brass; Writing—review & editing: Ian Apperly, Renée Baillargeon, Simon Baron-Cohen, Cristina Becchio, Harold Bekkering, Daniel Bernstein, Maxime Bertoux, Geoffrey Bird, Marcel Brass, Henryk Bukowski, Pascal Burgmer, Peter Carruthers, Caroline Catmur, Isabel Dziobek, Nicholas Epley, Thorsten Michael Erle, Chris Frith, Uta Frith, Carl Michael Galang, Vittorio Gallese, Delphine Grynberg, Francesca Happé, Masahiro Hirai, Sara D. Hodges, Philipp Kanske, Mariska Kret, Claus Lamm, Jean Louis Nandrino, Sukhvinder Obhi, Sally Olderbak, Josef Perner, François Quesque, Yves Rossetti, Dana Schneider, Matthias Schurz, Tobias Schuwerk, Natalie Sebanz, Simone Shamay-Tsoory, Giorgia Silani, Shannon Spaulding, Andrew R. Todd, Evan Westra, Dan Zahavi.

## Funding

## Competing interests

The authors declare no competing interests.
