## [Peer Review File · Communications Psychology]

4th Sep 23

Dear Dr Quesque,

Thank you for your patience during the editorial evaluation of your manuscript titled "Defining key concepts for mental state attribution: Towards a consensual lexicon" to Communications Psychology.

We have given the paper our careful consideration and find it of potentially great interest and utility. However, as discussed via email, your piece does not meet the criteria for a Research Article in Communications Psychology.

Instead, we think this piece could become a very widely read and important contribution to the field if you were prepared to have us consider it as a Comment. We therefore invite you to revise it in order to comply with our Comment format. Comments in the Nature Portfolio, including in Communications Psychology, are opinion pieces, not research contributions. Comments are peer-reviewed at the editors' discretion. Your Comment, with its strong focus on scientific a research question and theory, would be peer-reviewed by experts in the field. The peer-review files, including the editorial decision letters, reviewer comments, and author rebuttal letters will be included with the final article if your Comment is accepted for publication following peer review.

I have attached two documents to aid with revisions (tracked changes & clean manuscript). I provided detailed comments in the box, but in summary, we ask you to:

- 1) Move the Methods section to the Supplementary Information file: As this is no Research Article, the precise process you followed does not need to be presented in the main manuscript; instead it is sufficient to outline the process in broad terms in the manuscript (see end of first section) and focus your piece on your key argument/goal - achieving greater clarity and agreement in the use of mental state attribution terms.
- 2) Present your consensus lexicon in a box: Presenting the definitions as part of a display item not only gives them greater visibility, but also means that they do not count towards the word limit.
- 3) Revise the subheadings: According to our formatting guidelines, Comments do not have the sections usually associated with a Research Article; instead, the text should be structured by meaningful subheadings.
- 4) Move/revise Tables (see document for details)
- 5) Revise Figure 1 to make it more insightful (see document for details)
- 6) Revise references: Comments should only reference key sources that the readers need to understand a controversy or for critical further reading. Self-references (to any contributing author) should be entirely avoided. In particular, because the present manuscript is an opinion piece, there is limited scientific value in referencing the same opinion expressed previously by some of the same authors elsewhere.

7) Ensure the Comment is within word limit of 1600-1800 words

We hope you will be able to address these concerns before we send your manuscript for external review.

We shall hope to receive your revised version as soon as you are able to complete the suggested revisions. If something similar is published in the interim we will have to consider the impact it has on the novelty of a revised manuscript.

If you anticipate a delay of more than four weeks, please let us know. Should your manuscript be substantially delayed without notifying us in advance and your article is eventually published, the received date may be that of the revised, not the original, version.

If you are not interested in submitting a suitably revised manuscript in the future please let me know immediately so we can close your file. If you have any questions, please contact me.

Please use the link below when you are prepared to resubmit.

[link redacted]

Thank you for your interest in Communications Psychology.

Best regards,
Antonia Eisenkoeck

Antonia Eisenkoeck
Senior Editor
Communications Psychology

21st Dec 23

Dear Francois,

We apologise for the delay in processing your manuscript, as explained via email, we had experienced issues in the review process - thank you for your patience.

Your Comment titled "Defining key concepts for mental state attribution: Towards a consensual lexicon" has now been seen by 2 referees, whose comments appear below. In light of their advice, I am delighted to say that we are happy, in principle, to publish it in Communications Psychology under a Creative Commons 'CC BY' open access license.

We will not send your revised paper for further review if, in the editors' judgment, the referees' comments on the present version have been addressed. If the revised paper is in Communications Psychology format, in an accessible style, and of appropriate length, we shall accept it for publication immediately. I have attached an edited version of your manuscript, and ask you to attend to each comment in detail.

EDITORIAL REQUESTS:

*Reviewer #2 highlights that the piece reflects no true consensus, as the list of authors is not exhaustive. We ask that you undertake some edits to the piece, and in particular to the title, to reflect that it is a piece that reports agreement by experts, working towards a consensus, but that it is not, and cannot be representative of everyone's opinion.

*Relatedly, Reviewer #1 requests more information on how the panel was formed. This can be described in detail in the SI. We provide edits on your manuscript to help you navigate the balance between this valid request for transparency with the editorial requirement to present the piece as a Comment, not a piece that can be misread as a research Article.

*Reviewer #2 raises questions about the evidential support for some definitions. Please review the strength of your respective arguments in these instances.

* Please review the changes in the attached copy of your manuscript, which has been edited for style, and address the comments and queries I have added. If using Word, please use the 'track changes' feature to make the process of accepting your manuscript more efficient.

* Communications Psychology uses a transparent peer review system. On author request, confidential information and data can be removed from the published reviewer reports and rebuttal letters prior to publication. If you are concerned about the release of confidential data, please let us know specifically what information you would like to have removed. Please note that we cannot incorporate redactions for any other reasons.

*If you have not done so already, please alert me to any related manuscripts from your group that are under consideration or in press at other journals, or are being written up for submission to other journals (see www.nature.com/authors/editorial_policies/duplicate.html for details).

FORMATTING GUIDELINES:

You will find a complete list of formatting requirements following this link:

<https://www.nature.com/documents/commsj-style-formatting-checklist-comment.pdf>

Please use the checklist to prepare your manuscript for final submission. In the following, I also highlight some issues of particular importance.

** Title

Titles should be descriptive of the main message your manuscript conveys and should not exceed 90 characters (including spaces). Please note that punctuation is not allowed, nor are titles of the following format: "title: subtitle". Although the choice of title is largely yours, may I suggest the following:

'Defining key concepts for mental state attribution'? There are other suggestions in the attached file.

** Length

The ideal length for Comment article in Communications Psychology is 1,500 words. We have some flexibility, however, but please ensure that your text does not exceed 1,800 words.

** Main text

Please provide three or four section headings in the main text. These should relate to the content of the article rather than being generic. Headings should be no longer than 30 characters (including spaces) and should not use punctuation.

** Figures

Please remove all figures from the main text and upload them individually, one figure per file. To ensure the swift processing of your paper please provide the highest quality, vector format, versions of your images (.ai, .eps, .psd) where available. Text and labelling should be in a separate layer to enable editing during the production process. If vector files are not available then please supply the figures in whichever format they were compiled in and not saved as flat .jpeg or .TIFF files. If your artwork contains any photographic images, please ensure these are at least 300 dpi.

* Figures should be simple and informative — multi-part figures are best avoided.

* References

References appear as superscript Arabic numerals, in order of mention. The reference list mentions references in the numerical order in which they are mentioned in the main text. If a reference is cited more than once, the same number is used throughout the text and the reference receives a single entry in the reference list.

Only papers that have been published or accepted by a named publication should be in the reference list (preprints and citations of datasets are also permitted). Unpublished/Submitted research should not be included in the reference list; it should only be mentioned briefly and parenthetically in the main text. Note that no major arguments should rely on unpublished research.

Published conference abstracts and URLs for websites should be cited parenthetically in the text, not

in the reference list.

Footnotes are not used.

* Competing interests

Please include a "Competing interests" statement after the References. Note that we ask authors to declare both financial and non-financial competing interests. For more details, see <https://www.nature.com/authors/policies/competing.html>. If you have no financial or non-financial competing interests, please state so: "The authors declare no competing interests."

SUBMISSION INFORMATION:

* If you wish, you may also submit a visually arresting image, together with a concise legend, for consideration as a 'Hero Image' on our homepage. The file should be 1400x400 pixels and should be uploaded as 'Related Manuscript File'. In addition to our home page, we may also use this image (with credit) in other journal-specific promotional material.

* Your paper will be accompanied by a two-sentence editor's summary, of between 250-300 characters, when it is published on our homepage. We will use a variation of the preface for this purpose.

In order to accept your paper, we require the following:

* A cover letter describing your response to our editorial requests.

* A separate document detailing your point-by-point response to any issues raised by our referees (please include the referees' comments in this document).

* The final version of your text as a Word or TeX/LaTeX file, with any tables prepared using the Table menu in Word or the table environment in TeX/LaTeX and using the 'track changes' feature in Word.

* Production-quality versions of all figures, supplied as separate files. Photographic images should be 300 dpi in RGB format (.jpg, TIFF or native Photoshop format) and any labels/scale bars included in a separate layer from the image. Line art, graphs and schemes should be vector format (.ai, .eps, .pdf); Adobe Illustrator files are preferred and will minimize production time. Any chemical structures or schemes contained within figures should additionally be supplied as separate Chemdraw (.cdx) files.

At acceptance, the corresponding author will be required to complete an Open Access Licence to Publish on behalf of all authors, declare that all required third-party permissions have been obtained.

Please note that your paper cannot be sent for typesetting to our production team until we have received this information; **therefore, please ensure that you have this ready when submitting the final version of your manuscript.**

ORCID

Communications Psychology is committed to improving transparency in authorship. As part of our efforts in this direction, we are now requesting that all authors identified as 'corresponding author' create and link their Open Researcher and Contributor Identifier (ORCID) with their account on the Manuscript Tracking System (MTS) prior to acceptance. ORCID helps the scientific community achieve unambiguous attribution of all scholarly contributions. For more information please visit <http://www.springernature.com/orcid>

For all corresponding authors listed on the manuscript, please follow the instructions in the link below to link your ORCID to your account on our MTS before submitting the final version of the manuscript. If you do not yet have an ORCID you will be able to create one in minutes.

IMPORTANT: All authors identified as 'corresponding author' on the manuscript must follow these instructions. Non-corresponding authors do not have to link their ORCIDs but are encouraged to do so. Please note that it will not be possible to add/modify ORCIDs at proof. Thus, if they wish to have their ORCID added to the paper they must also follow the above procedure prior to acceptance.

To support ORCID's aims, we only allow a single ORCID identifier to be attached to one account. If you have any issues attaching an ORCID identifier to your MTS account, please contact the Platform Support Helpdesk.

[link redacted]

We hope to hear from you within two weeks; please let us know if the process may take longer.

Best regards,

Marike, on behalf of Antonia Eisenkoeck

Marike Schiffer
Chief Editor
Communications Psychology

Antonia Eisenkoeck
Senior Editor
Communications Psychology

REVIEWERS' EXPERTISE:

Reviewer #1: social cognition, social dysfunction

Reviewer #2: social cognition

REVIEWERS' COMMENTS:

Reviewer #1 (Remarks to the Author):

Here, the authors tackle a nagging problem within the social cognitive literature and attempt to provide some clarity regarding the many terms that are used to describe processes related to inferring the mental states of others. Via expert consensus, the authors identified the most widely used and accepted terms and provided definitions of these terms that can be used to differentiate them and to provide greater specificity in future work.

The paper is well done and timely. As noted by the authors, there has been a proliferation of terms in the literature that has contributed to confusion and a lack of scientific rigor. The results of the expert consensus process are clear, and if adopted by the scientific community, will make a significant and positive impact. I have only two small suggestions that I believe will strengthen the paper, and both should be easy to address via minimal revision. I'd like to thank the authors for taking on this task and for the thoughtfulness of this work.

Minor Suggestions:

1. In the introduction, the authors rightfully note that heterogeneity in terminology can lead to problems in comparability of findings and may obscure methodological variability. These are important points, and it may be helpful to readers to underscore them with concrete examples.
2. With the understanding that the present manuscript is submitted as a "Comment" paper and that word limits are likely tight, I would still like to recommend that a brief summary of the composition of the expert panel be included in the main paper. While this info is provided in the supplement, the diversity of the panel in terms of location, field of study, and stage of career is a real strength of the current work that lends credence to the results.

Amy Pinkham

Please note that I generally sign my reviews in favor of full transparency.

Reviewer #3 (Remarks to the Author):

Quesque and colleagues provide a selection of consensus definitions concepts in the field of mental state attribution, which emerged from discussions by an expert panel, and subsequent validation by members of this panel.

I find the undertaking extremely valuable, and applaud the authors for doing this. I also very much agree with all (well, mostly -- see below) of the recommendations and suggestions for new replacement for older terms. The manuscript is also very nicely written and to the point. However, I have one major concern, and several minor ones.

My major concern is how the panel of experts was selected. If I understand correctly, the panel

consisted initially of the members of a workshop who then suggested other researchers to "exhaustively" sample the field (62 in total, a third of which have contributed). I am not convinced at all that this is a good representation of the field, and neither that it is "exhaustive". Especially given how scientific networks are organised, it is very unlikely that the researchers were therefore able to go beyond their own "bubble" -- and of course the whole field is of course much, much wider than the 40 contributors. Looking at the author list, I can think of several important researchers who have done seminal work in perspective taking, emotion perception, or children's theory of mind, which do not seem to have been included. It also does not include many early career researchers for whom the authors claim to speak. Especially given recent calls for a more democratic science that goes beyond cliques I would have looked for a much more principles approach, perhaps based on core search terms similar to how it is used for systematic reviews. Especially given recent developments in online research, it would have been easy for such a broader panel of experts to be sampled, which would have allowed for the development of a true consensual lexicon that is shared across researchers in the field.

Minor/more specific points:

Box 1, definition of "mentalizing". Mentalizing, to this reviewer, goes beyond the ability of attributing mental states to others but also involves the ability to reason about mental states (e.g., how one mental state can follow from another), how mental states are driven by person-external influences (e.g., an emotional situation one finds the other person in), can be read from behavior and used to predict future behaviour. Does their definition include these features? It seems that such a clarification is necessary in particular because the authors claim further down that Theory of Mind is one way to mentalize, but here its definition explicitly seems to include at least some of the features as part of the heuristics ("mental states are correlated with behaviors").

Box 1, definition of Empathy. Their definition ("the ability to experience others' affective states") should perhaps read the "ability to experience others' (inferred) affective states", as we cannot be certain which affective state the other person truly has (it is hidden as the authors note at the start, and it needs to be inferred from behaviour, context etc.).

Box 1, definition of Perspective taking. I find the following sentence very unclear: "Importantly, perspective-taking can refer to a spatial and a temporal perspective (i.e., evaluating my current behavior based on a potential future perspective)" What does spatial and temporal perspective mean here? Maybe an example is needed for each of these perspectives.

Page 11, top: I found this section almost impenetrable: "For example, "empathic perspective-taking", which is classically defined as 'imagining how a target person feels', appeared to be conceptually closer to our proposed definition of mentalizing than to those obtained for empathy (as its definition does not imply experiencing the emotional state of the targeted person) or perspective-taking (as its definition does not imply the adoption of the other perspective)"

Can this be put more simply? Also I was puzzled that, according to the authors, "imagining how a target feeling feels" does not seem to imply "to experience others' affective states,"? For me, this distinction between imagining and experiencing feels incredibly subtle, especially given that we do not have 1st hand experience of someone else's affective states and have to infer it (which is very close to imagining). I am therefore not convinced that this subtle distinction would be underwritten by the majority of researchers. Can the author clarify whether this distinction itself is really consensual and how it was tested? I am also not convinced that empathic perspective taking can be equated with "mentalizing about affective states". Empathic perspective taking to me implies (or at

least can imply) an emotional experience, but mentalizing about affective states does not. It would be good if these points could be clarified.

In this document, we provide below (in bold) a point-by-point response to the referees' comments.

Reviewer #1

Here, the authors tackle a nagging problem within the social cognitive literature and attempt to provide some clarity regarding the many terms that are used to describe processes related to inferring the mental states of others. Via expert consensus, the authors identified the most widely used and accepted terms and provided definitions of these terms that can be used to differentiate them and to provide greater specificity in future work.

The paper is well done and timely. As noted by the authors, there has been a proliferation of terms in the literature that has contributed to confusion and a lack of scientific rigor. The results of the expert consensus process are clear, and if adopted by the scientific community, will make a significant and positive impact. I have only two small suggestions that I believe will strengthen the paper, and both should be easy to address via minimal revision. I'd like to thank the authors for taking on this task and for the thoughtfulness of this work.

We wish to thank the referee for this positive feedback.

1. In the introduction, the authors rightfully note that heterogeneity in terminology can lead to problems in comparability of findings and may obscure methodological variability. These are important points, and it may be helpful to readers to underscore them with concrete examples.

In the new version of the manuscript, we amended the introduction in order to include concrete examples (e.g. P.5. « Interestingly, this lack of consensus is observed not only across different disciplines but also within certain disciplines in which umbrella terms at the conceptual level cover methodological variability (e.g. the same concept will be used to refer to explicit beliefs ascription and emotional faces categorization). This heterogeneity leads to problems in the comparability and generalization of findings. »)

2. With the understanding that the present manuscript is submitted as a "Comment" paper and that word limits are likely tight, I would still like to recommend that a brief summary of the composition of the expert panel be included in the main paper. While this info is provided in the supplement, the diversity of the panel in terms of location, field of study, and stage of career is a real strength of the current work that lends credence to the results.

In the new version of the manuscript, and also based on reviewer's 2 comment regarding the panel selection, we amended the introduction in order to briefly summarize the composition and formation of the expert panel (e.g. P.6. « To initiate the development of a shared lexicon, FQ and MB engaged with experts in the field. Although the selection was not exhaustive, it encompasses a diverse array of disciplines, countries, and career stages. Then, our panel of researchers collaboratively identified essential components of such definitions for each construct. In this process, we tried to avoid convergent use of terms (different terms referring to the same concept) and divergent use of terms (one term referring to different concepts). Importantly, such a pruning should be as consensual as possible to ensure that researchers commit to the outcome of this process. Here we present the product of this collaborative project – a common lexicon of mental state attribution terms developed in consultation with leading researchers from diverse relevant research fields (e.g. affective neuroscience, philosophy, social psychology).»)

Reviewer #2

Quesque and colleagues provide a selection of consensus definitions concepts in the field of mental state attribution, which emerged from discussions by an expert panel, and subsequent validation by members of this panel.

I find the undertaking extremely valuable, and applaud the authors for doing this. I also very much agree with all (well, mostly -- see below) of the recommendations and suggestions for new replacement for older terms. The manuscript is also very nicely written and to the point. However, I have one major concern, and several minor ones.

We wish to thank the referee for this positive feedback.

My major concern is how the panel of experts was selected. If I understand correctly, the panel consisted initially of the members of a workshop who then suggested other researchers to "exhaustively" sample the field (62 in total, a third of which have contributed). I am not convinced at all that this is a good representation of the field, and neither that it is "exhaustive". Especially given how scientific networks are organised, it is very unlikely that the researchers were therefore able to go beyond their own "bubble" -- and of course the whole field is of course much, much wider than the 40 contributors. Looking at the author list, I can think of several important researchers who have done seminal work in perspective taking, emotion perception, or children's theory of mind, which do not seem to have been included. It also does not include many early career researchers for whom the authors claim to speak. Especially given recent calls for a more democratic science that goes beyond cliques I would have looked for a much more principles approach, perhaps based on core search terms similar to how it is used for systematic reviews. Especially given recent developments in online research, it would have been easy for such a broader panel of experts to be sampled, which would have allowed for the development of a true consensual lexicon that is shared across researchers in the field.

We wish to thank the referee for this important remark. We acknowledge that the common lexicon generated by the 45 final contributors could not reflect a real cross-disciplinary scientific consensus. In order to account for the different limits listed above by the referee, the revised version of the manuscript now:

- Explicitly mentions that the methodology employed may have limited the inclusion of early career researchers (e.g. P.6.: « To initiate the development of a shared lexicon, FQ and MB engaged with experts in the field. Although the selection was not exhaustive, it encompasses a diverse array of disciplines, countries, and career stages. (...)»).
- Indicates that this project can only represent a first step, and that a wider consultation will be necessary (e.g. P.10.: « The collaborative approach to develop a common lexicon proposed here constitutes a crucial first step towards better practices. This approach also has limitations that we want to address. First, despite a willingness to involve as many experts as possible, we acknowledge that our consortium might be biased and therefore it would be crucial to expand it in the next steps. Thus, we think that this initiative will have to be followed by other collective initiatives to ensure representation of different cultures, as well as the different aspects of psychology and social neuroscience. »).
- Largely limits the use of the term « consensus » (i.e. We (1) changed the title of the manuscript (2), updated the preface, and (3) amended the main text in order to limit the use of the term « consensus » (e.g. p.7 « Shared lexicon », p.11 « Advantage of a common lexicon », see ms for all instances).

Minor/more specific points:

Regarding the next three minor points listed below by the reviewer, we understand that they might have a different opinion about some of the terms. This was also the case with some of the experts included in the panel for all terms. Accordingly, we do not have a 100 % agreement on any of the definitions. However, the opinion of the reviewer only reflects one opinion and it would be problematic to give their opinion a stronger weight than that of the authors given the very nature of the project. We still address these three points below and explicit the underlying reasoning which led to these definitions.

Box 1, definition of "mentalizing". Mentalizing, to this reviewer, goes beyond the ability of attributing mental states to others but also involves the ability to reason about mental states (e.g., how one mental state can follow from another), how mental states are driven by person-external influences (e.g., an emotional situation one finds the other person in), can be read from behavior and used to predict future behaviour. Does their definition include these features? It seems that such a clarification is necessary in particular because the authors claim further down that Theory of Mind is one way to mentalize, but here its definition explicitly seems to include at least some of the features as part of the heuristics ("mental states are correlated with behaviors").

These features appeared in some definitions of our panel but were finally judged to be irrelevant as essential parts of the definition when we confronted the different perspectives within our group. This reasoning was supported by the fact that we consensually voted for "mentalizing" to be the term used "by default" when addressing the ability to attribute mental states – independently of the cognitive processes involved. In order to allow this recommended use, we believe that the definition of "mentalizing" should not include elements regarding how it works, neither regarding its purpose. Doing so would inevitably lead some researchers to introduce alternatives as generic terms if their conception does not fit with our present definition.

Box 1, definition of Empathy. Their definition ("the ability to experience others' affective states") should perhaps read the "ability to experience others' (inferred) affective states", as we cannot be certain which affective state the other person truly has (it is hidden as the authors note at the start, and it needs to be inferred from behaviour, context etc.).

We understand the referee's concern but we believe that this reasoning (i.e. "mental states are hidden and can only be inferred") could be applied to all our definitions. We acknowledge this point from the very beginning of the manuscript, as the reviewer noticed. The nature (or even existence) of the inference is an important subject of debate among neuroscientists, psychologists and philosophers. For this reason, we chose to not address this debate in this article as this would largely exceed our present purpose (i.e. have common definitions).

Box 1, definition of Perspective taking. I find the following sentence very unclear: "Importantly, perspective-taking can refer to a spatial and a temporal perspective (i.e., evaluating my current behavior based on a potential future perspective)" What does spatial and temporal perspective mean here? Maybe an example is needed for each of these perspectives.

We wish to thank the referee for this suggestion. Accordingly, the revised version of the manuscript now includes two examples. (see box 1: « We recommend defining perspective-taking as "the process by which one represents others' mental states, by adopting their perspective". Based on this definition, perspective taking would refer to a specific form of mentalizing. Importantly, perspective-taking can refer to a spatial (e.g., "my colleague think that I am working as they can't see my screen") and a temporal perspective (e.g., "tomorrow my colleague will be

disappointed, if I don't prepare the meeting now"). 82,5% of our panel agreed with this definition. »)

Page 11, top: I found this section almost impenetrable: "For example, "empathic perspective-taking", which is classically defined as 'imagining how a target person feels', appeared to be conceptually closer to our proposed definition of mentalizing than to those obtained for empathy (as its definition does not imply experiencing the emotional state of the targeted person) or perspective-taking (as its definition does not imply the adoption of the other perspective)"

Can this be put more simply? Also I was puzzled that, according to the authors, "imagining how a target feeling feels" does not seem to imply "to experience others' affective states,"? For me, this distinction between imagining and experiencing feels incredibly subtle, especially given that we do not have 1st hand experience of someone else's affective states and have to infer it (which is very close to imagining). I am therefore not convinced that this subtle distinction would be underwritten by the majority of researchers. Can the author clarify whether this distinction itself is really consensual and how it was tested? I am also not convinced that empathic perspective taking can be equated with "mentalizing about affective states". Empathic perspective taking to me implies (or at least can imply) an emotional experience, but mentalizing about affective states does not. It would be good if these points could be clarified.

We understand the referee's concern and revised this section accordingly, now explicitly mentioning that the meaning of each discarded terms was based on our panel's representation of these terms (see p.7: « Table 1 summarizes our recommendations for each of the discarded terms. It might appear surprising that for some terms (e.g., "empathic perspective-taking") we recommend favoring alternatives which, at first sight, might seem less lexically related (e.g., "mentalizing about affective states") than others (e.g., "empathy" or "perspective-taking"). The reason driving these decisions is that we focused on the conceptual idea underlying the panel's definitions of these terms, rather than on lexical proximity or alternative definitions. In this case, as "empathic perspective-taking", was defined as 'imagining how a target person feels', it appeared to be conceptually closer to our proposed definition of mentalizing than to those obtained for empathy (as its definition does not imply experiencing the emotional state of the targeted person) or perspective-taking (as its definition does not imply the adoption of the other perspective). »). We acknowledge that other researchers might have a distinct meaning for any discarded terms. If so, they should legitimately favor other alternatives.